# China’s Elder Care Policies 1994–2020: A Narrative Document Analysis

**DOI:** 10.3390/ijerph19106141

**Published:** 2022-05-18

**Authors:** Marion F. Krings, Jeroen D. H. van Wijngaarden, Shasha Yuan, Robbert Huijsman

**Affiliations:** 1Department of Health Policy and Management, Erasmus University Rotterdam, P.O. Box 738, 3000 DR Rotterdam, The Netherlands; krings@eshpm.eur.nl (M.F.K.); vanwijngaarden@eshpm.eur.nl (J.D.H.v.W.); huijsman@eshpm.eur.nl (R.H.); 2Institute of Medical Information & Library, Chinese Academy of Medical Sciences & Peking Union Medical College, Beijing 100020, China

**Keywords:** aging policy, elder care, policy analysis, qualitative research, China

## Abstract

Until the 1980s, institutional elder care was virtually unknown in China. In a few decades, China had to construct a universal social safety net and assure basic elderly care. China’s government has been facing several challenges: the eroding traditional family care, the funding to assure care services for the older population, as well as the shortage of care delivery services and nursing staff. This paper examines China’s Five-Year Policy Plans from 1994 to 2020. Our narrative review analysis focuses on six main topics revealed in these policies: care infrastructure, community involvement, home-based care, filial piety, active aging and elder industry. Based on this analysis, we identified several successive and often simultaneously strategic steps that China introduced to contend with the aging challenge. In Western countries, elder care policies have been shifting to the home care approach. China introduced home care as the elder care cornerstone and encouraged the revival of the filial piety tradition. Although China has a unique approach, the care policies for the aged population in China and Western countries are converging by emphasizing home-based care, informal care and healthy aging.

## 1. Introduction

Worldwide aging is considerable and in particular in Europe and Asia. The percentage of the population of Germany above 65 reached 22% in 2020, that of France reached 21%, and that of China reached 14%. Although China’s percentage is lower, the challenge is more severe, given the size of the country with an elder population over the age of 65 reaching 191 million [1]. Moreover, China’s aging process is very fast with an expected pronounced acceleration between 2015 and 2040. By 2050, about 26% of China’s total population will be 65 or older, and 8% will reach 80 years and older [2]. At the same time, until the 1980s, institutional elder care was virtually unknown in China, and the few facilities that existed were social welfare institutions run by the government [3]. China had to build an elder care system basically from scratch in only a few decades. Nonetheless, studies about population aging and elder policies tend to focus mainly on Western countries [4].

Kudo et al. [5] argue that while developing nations face socioeconomic challenges similar to those of developed nations, they moreover face the pressure to meet the basic needs of the growing aging population. China was confronted with the task of building not only an elderly care infrastructure but also a pension and social insurance system while facing a shrinking labor force in the near future. Massive migration (1979–2015) to urban areas, especially of young people, combined with the one-child policy, left many older individuals in rural communities without traditional family support (filial piety) [6,7,8,9]. This problem required institutional interventions. To meet these challenges, the Chinese government launched a first perennial policy for the aging undertaking in 1994. This initial seven-year plan was followed with subsequent five-year policy plans. Many countries introduced policies and innovative strategies to deal with aging [8]. The paradigms of active and healthy aging emerged worldwide as a policy response to reduce the growing financial impact of health care and to contain the increasing overall challenges of population aging [10]. It is generally recognized that older individuals can contribute and provide meaningful economic and social benefits, especially when active and healthy [11]. Moreover, to prevent social isolation and loneliness and related health care issues, it is vital for elder individuals to continue to be active members of society.

Notwithstanding the fact that socioeconomic changes and subsequent policy responses vary across continents, almost all countries are facing similar social and political pressures that influence both the demand and supply of elderly care [12]. Today, there is a tendency of policies to shift from government-centered to more neighborhood and home-centered approaches in several countries [13]. For the setup of the elderly care system, China could follow the example of Western models or choose the Asian approach of community care or, alternatively, create its own model [14].

This paper aims to provide the first in-depth analysis of China’s five successive Five-Year Policy plans for the aging undertaken between 1994 and 2020. This research investigates how these elderly policies have evolved within China’s specific socioeconomic environment in order to face the aging challenges at the national, community and individual levels.

### Background

China’s State Council and relevant ministries are in charge of overall policy planning, promulgating measures and defining the standards for elderly care services, facilities and the care workforce. These policies are implemented by provinces and autonomous regions to suit local political governance [15,16]. The institutional interrelationships are characterized as politically centralized and economically decentralized [17]. Each sub-government plays a significant role in financing and enjoys a rather autonomous economic position controlling its own infrastructure, industries, and markets. The Community Committees, China’s lowest administrative level, are in charge of the administration. The notion of community as home-proximate care for the elderly has evolved in a variety of political and social environments around the world [18].

In the 1950s–1970s, the vast majority of Chinese had fair access to essential health care through government insurance (mainly for civil servants) and workers’ health insurance schemes in urban areas and cooperative medical insurance in rural areas [19,20]. China’s social safety net fell apart at the start of the 1978 reforms [21]. To protect family safety, the Chinese Constitution adopted in 1982 article 49: ‘Parents have the duty to rear and educate their minor children, and children who have come of age have the duty to support and assist their parents’ [22].

In 2011, over 95% of the Chinese population was covered by health insurance [23]. In 2014, China’s global GDP increased to 16.6%, surpassing that of the US (16.0%) and second to that of the EU (17.0%) [24]. Although after the economic reform in 1978, China’s economic development was blooming, it took the country over thirty years to improve the social safety net. In this time span, China lifted over half a billion people out of poverty [24]. 

The Chinese legal retirement age is 60 years for men and 55 years for women. Currently, China is considering extending the statutory retirement age (14th Economic and Social Five-Year Plan 2020) [25]. In 2017, the basic pension insurance participation rate reached 90%, and the basic medical insurance participation rate was over 95% [26,27]. However, China’s basic old-age pension remains inadequate. In approximately 2015, the basic pension amounted to only nine US dollars monthly [20]. Notwithstanding the large increase in income inequality, much of China’s population has experienced real income rising [24]. Pension and social insurances received little attention in elderly care policies. This can be explained by the fact that social insurances are comprehensively considered in China’s Five-Year National Economic and Social Development planning. However, the elderly policy plans stress the importance of social insurances and the safety net in general.

After having comprehensively guaranteed a basic nationwide health insurance coverage, China is today facing the challenge of conceiving and implementing an appropriate elder care delivery system.

## 2. Materials and Methods

### 2.1. Design and Setting

Although China’s population is aging fast, China’s policy plans regarding the older population have not caught much attention over the past years. Some authors referred to the 12th or 13th Five-Year policy plans. Very little has been written about the impact of these policy plans, and we therefore decided to fill this gap. We developed a narrative review of the governmental documents and secondary literature, using search terms in English and Chinese to perform a topic-lead explorative analysis of China’s national aging policy plans since 1994.

### 2.2. Data Collection

China’s Five-Year policy plans (1994–2020) on the aging undertaking are our core data. Meanwhile, secondary literature is used to understand the history, context related to these policies. The two data sources are essentially obtained by means of desktop research:China’s official publicly available policy documents (in Chinese);Official Chinese databases: China Statistical Yearbook (in English), China’s State Council (on the Internet);Reports of international organizations (World Health Organization, World Bank);Secondary literature: search for international papers on Google Scholar, university libraries of Erasmus university and organizations worldwide.

### 2.3. Data Analysis

The sources in the Chinese language have been translated, where necessary, by the first authors and verified by a native speaker. For the qualitative analysis of the policies, the first author went in detail through China’s elder Five-Year Policy Plans 1994–2020 to detect the main themes and their frequency of appearance (counting the paragraphs) in the policies. These were checked by and discussed with the second and last authors. After several discussions and a literature search for related concepts and developments, the topics were grouped into six fields: care infrastructure, community involvement, home-based care, filial piety, active aging and elderly industry. The findings are visualized in Figure 1, which provides an insight of the frequency of the themes appearing throughout these policies.

Secondary literature was collected to deepen our understanding of each theme in the Chinese context. During each step, the findings were discussed between all authors until consensus was reached. Finally, the findings were checked by two experts on Chinese elderly policy. In the analysis of China’s successive elderly care policies 1994–2020, we discovered a line of steady development of the policy measures.

## 3. Results

### 3.1. Overview of the Five Chinese Elderly Policy Plans from 1994 until 2020

(a)Figure 1 provides an insight of the frequency of attention of the six themes appearing in these policies in the different time-spans.

We identified three distinct periods of elderly care development. The first development period 1994–2001 includes the Seven-Year Plan and the 10th Five-Year Plan. Economic development was still considered to be the solution to the improvement of the aging population conditions. Focus was on the construction of elderly homes. 

The 11th Five-Year Plan, 2006–2010, represents the second development period. This FYP shows a deeper-going social inclination with the aim of constructing a harmonious society. In 2009, a major nation-wide health care reform was inaugurated. This second period is a transitional period. 

The third period, 2011–2020, comprises the 12th and 13th Five-Year Plans. In the 12th FYP, China’s central government officially proclaims a new elderly care infrastructure: ‘elderly home care as foundation; community elderly care as support; and state institutional care as supplement’. This period enhances the traditional virtue of filial piety. The switch from elderly homes in the first period to home care more recently resembles the development of Western elderly care. In several European countries, the focus on home care was intensified in the 2000s [12]. 

(b)Overview of China’s elderly policy plans 1994–2020.

Table 1 presents a chronological overview of the five Chinese elderly policy plans 1994–2020. The table clearly indicates a summary of the substance of the examined policies and mentions the achievements of each policy plan. 

The table serves as a framework while going through the details of the six themes in the following sections.

### 3.2. Elderly Care Service Infrastructure

At the launch of the first perennial elderly policy plan in 1994, the nursing system in China was rudimentary. Delivery of medical care services relies heavily on the public hospital sector. Public hospitals represented 34.7% of all hospitals and 72.5% of all beds in 2019 [28]. Because government funding for health dramatically decreased since the 1980s, hospitals must find their own budgets to cover expenses. Health providers started to overprescribe drugs and treatments [6]. Out-of-pocket payments soared, and patients avoided specialized health care [29]. Rural elderly individuals were affected the most by these changes [30]. The major health care reform in 2009 started to improve the overall system [31]. A decade later, important progress was the decrease in out-of-pocket health expenses from 59% in 2000 to 29% in 2017 [32].

In the policies from 1994 to 2005, the government paid more attention to institutional homes [33]. The goal of 30 beds per 1000 elderly individuals was set within the 12th Five-Year Policy Plan (12th FYP hereafter). A report issued by the Peking Union Medical College states almost 1.6 million beds remained vacant by the end of 2016 [34]. Apparently, the new facilities are often too remote from the original living community. Luo and Zhan (2011) argue that older adults, when not fully cared for by their children, suffer from personal stigmatization [35]. 

The 13th Five-Year Policy Plan (13th FYP hereafter) urges the multiplication of rehabilitation departments, hospice care institutions and geriatric training and research. To ensure funding, the government encouraged in 2016 the private sector to run 50% of China’s elderly care facilities and beds [36]. China targeted employing 6 million nurses in elderly care establishments by 2020, a growth with approximately 1.5 million. The 11th Five-Year Policy Plan (11th FYP hereafter) encourages the training of community nurses and General Practitioners (GPs). The shortage of GPs remains a key issue. In 2019, the country counted a total of 365,082 GPs, representing 2.6 GPs per 10,000 people [37]. The government started to train 500,000 additional GPs in 2018 to reach a rate of 5 GPs per 10,000 residents [38]. People still prefer to go to hospitals even for minor consultations, which leads to long waiting times in hospitals. China now focuses on the support of information technology and has started the establishment of 68 internet hospitals in 2017 [29]. 

Internet platforms and remote monitoring at the community and home care levels have been set up in major cities [39]. China’s overall elderly care infrastructure is developing rather fast, yet manpower resources are only slowly increasing. Since the efforts of the state to provide elderly care homes have not succeeded, more attention is given to community and home care. The introduction of internet information platforms and virtual nursing can attenuate the shortage problem of nurses and GPs.

### 3.3. Community Care Involvement

Community care covers three main functions: community day care, home-based care support as well as cultural and sport activities. Community involvement is encouraged in all policy plans. To develop capacity, the central government issues a range of policy measures to attract and stimulate service providers to enter the elderly care market [40]. First, in 2018, the government abolished the mandatory need for a license to establish a community care institution. 

Elderly care policy plans request local governments to set up facilities for senior education and leisure (parks, green spaces, sports, and recreational sites) to support healthy aging. In 2006, the central government announced the national availability of 30,000 sport grounds and over 670,000 recreational amenities for the older population [41]. Cultural activities are highly recommended in policies. Today, China has approximately 70,000 senior colleges providing courses for over 8 million participants [42].

To implement the facilities, local governments face significant financial investments. Decentralized economic development has given rise to socioeconomic disparities. While provincial capitals or large cities possess many facilities, infrastructure remains scarce in underprivileged rural areas [4,39]. More recently, efforts have been made to develop community services in rural areas. Communities play an important role in the organization of outdoor and indoor health, leisure facilities as well as in the delivery of home care for the elderly.

### 3.4. Home-Based Care

China’s elderly care is often described as a ‘9073’: 90% of elderly are cared for by family, 7% receive community care and 3% live in a nursing home [40]. According to Chinese tradition, adult children are expected to take care of their aging parents. Traditionally, only childless and impoverished elder individuals entered public homes [33] and Chinese elderly can therefore feel a prejudice toward entering a nursing home. More recently, the policies are stimulating home care. The 12th FYP announces the new mode: home-based care represents the cornerstone, community care is the backing and institutions are supplements. The 12th and 13th FYPs moreover incite family members to live close to their parents and promote intergenerational cohabitation. Long-term care becomes crucial when older individuals encounter difficulties in conducting their daily activities due to disability [43]. The family doctor contracting service implemented in 2016 further stimulates integrated care for the home-based older population, which is one of the priority groups to be served by family doctors [44].

Recognizing that technological progress can facilitate home care, the government is promoting smart digital technology [40]. Virtual nursing systems, robotics, monitoring and call-service platforms are more widely introduced [39]. From 2015 to 2020, the central government allocated 5 billion yuan (approximately USD 743 million) to encourage new pilot programs of home-based elderly care services to be provided by communities in 203 cities [45]. China’s elderly individuals prefer home-based care and are assisted by community services when needed. Over the past decade, home care has become a priority in the policies, and China focuses on the development of smart home care technology.

### 3.5. Filial Piety

Throughout history, Chinese people have carried on the tradition of family support and respect for older people [46]. In 1996, China enacted the Elder Rights and Protection Law to mandate that adult children take on a moral obligation for their parents’ care, including the financial responsibility to provide for their medical needs and decent housing [47]. While material support matters, respect and affection are more highly valued [48,49,50]. The Chinese family is generally considered as the minimum social unit, not the individual [46]. Filial piety affirms the norms within the family and provides ethical foundations for the social order. Children’s ‘filial demonstration’ toward parents is socially judged as honorable [49,51]. 

The 12th FYP encourages traditional values, and states that outstanding filial piety behavior should be rewarded. Filial piety belongs to the core doctrine of the Confucian teachings [47], and it is important to remind that the 12th FYP was published after the reestablishment of Confucian teachings by China’s government in 2007. This policy advocates mutual affection as well as a warm and harmonious family atmosphere. The 13th FYP stipulates that elderly care must be ‘incorporated into the social morality and family virtues to create civilized communities’. The 12th and 13th FYPs request adult children to assume their filial piety commitment. In contrast to Western cultures, the identity of a ‘person’ is traditionally not rights-oriented but duty-oriented [52]. However, the effects of the one-child policy (abolished in 2016), demographic changes and the growing participation of Chinese women in the labor market have weakened the traditional caregiving structure [53]. 

To restore China’s demography, couples are now allowed to have a third child [54]. Parent–child interrelationships are slowly becoming more egalitarian. Children often engage in far-reaching careers or even going abroad [9], which implies living far away from their parents. Gui and Koropeckyj-Cox (2016) [55] point out that trust in filial piety nevertheless remains strong in people’s lives, especially among older individuals. China’s elderly population is greatly dependent on informal care provided by family members. The efforts of the state over the past years to reinforce traditional family values appear to have consolidated the virtue of filial piety.

### 3.6. Active and Healthy Aging: ‘To Add Life to the Years That Have Been Added to Life’ (United Nations 1999)

Good health and longevity is one of the most respectful greetings one can convey to a Chinese adult [56]. The first policies emphasize active aging. The ‘young and healthy elderly’ are encouraged to participate in social development (Seven-Year Policy Plan, 1994). This approach can be explained by the fact that in the mid-1990s, competitive pressure pushed state-owned enterprises to lay off workers. Men were often laid off at the age of approximately 50 and women 40. Over 30 million of these workers were laid off [57].

The policies show concern for elderly and the urge to enrich their spiritual and cultural lives. Spiritual well-being is highly cherished in traditional Chinese culture [58]. The policy plans promote self-reliance. Elderly are urged to participate in social events and to volunteer and engage in economic activities, i.e., to set up their own businesses (13th FYP). The importance of physical exercises for the elderly is mentioned repeatedly. Over 50% of the elderly should be engaging in physical fitness (12th FYP).

In Western countries, active and successful aging is associated with the ability to live independently. The Chinese ‘cultivating and nurturing of the self’ is not associated with this individualistic approach of the West but with an interdependent self, which is determined by one’s place in society and family structure. The willingness of families to care for the elderly is perceived as a sign of successful aging [3]. Healthy aging appears in the 12th and 13th FYPs. Following the international trend, China published the blueprint of Healthy Aging in 2016 [59], placing it in the Confucian tradition: to cultivate the moral self and to conserve one’s vital powers. ‘Healthy aging’ is not merely intended to provide older people with a happy lifetime; the individual health care responsibility is related to the intention of civic duty that urges individuals to stay healthy and avoid becoming the state’s burden [51].

### 3.7. Elderly Care Industry

The elderly care industry is developing. The 11th FYP advises ‘guiding the elderly to rational consumption’ and simultaneously ‘cultivating the consumer market for senior products and services’. The 13th FYP announces that the elderly service market will be ‘fully liberalized and market players will be supported’. The 12th and 13th FYPs encourage the establishment of call-service platforms and virtual nursing systems. The new generation of digital technology, cloud computing, artificial intelligence and robotics is developing fast. In 2015, the central government launched the Guiding Opinions of Actively Promoting Internet + Actions, which was followed by ‘The Action Plan for the Development of Smart Health and the Elder Care Industry (2017–2020)’ in 2017. A total of 99 companies were selected in the course of 2017 and 2018 to enter the smart technology market with a focus on home and community care services [36].

China’s ‘Big Health Industry’ includes health and medical care sectors. The elderly care sector comprises 33% of the total market size, covering a broad range of products, such as care services, facilities, health equipment, smart technology, insurance and tourism. The elderly care industry also comprises real estate. Parallel to the modest set-up of elder homes, luxury elder real estate is developing for the well-off in numerous urban areas [60]. The market for China’s elder care products is growing fast. In 2017, the market value was estimated at approximately US$0.79 billion, which accounts for a 12% increase over 2016. The prediction for the 2030 amount is US$15.22 billion [36].

Commercial health and LTC insurance are part of the market. Over 2400 insurance products are available for people aged 65 and above [61], and approximately 59 million elderly are participating in 2020 in the commercial health insurance supplement [61]. For long-term care (LTC), the Chinese government is guiding approximately 49 pilots in 2020 [61] with virtual platforms for common elderly care services. Since 2011, policies have mentioned the need for LTC. The LTC insurance market is dominated by state enterprises and private conglomerates [62]. Small enterprises cannot easily survive, since the sector requires material long-term investments offering a low (but steady) return [63]. Major Chinese enterprises are expanding internationally, such as the conglomerate China Oceanwide, which acquired Genworth in 2016, the largest LTC insurance company in the United States [62]. China’s young LCT insurance branch can benefit from these experiences. Products specializing in the older population are becoming a major industry.

## 4. Discussion

China is confronted with the largest aging population of the world. Although after the economic reform in 1978, China’s economic development was blooming, it took the country over thirty years to improve the social safety net. At the time of the first policy in 1994, elderly care infrastructure hardly existed. China had to develop a whole system de novo, as occurred before in many European countries around the 1950s. In 1994, China promulgated the first perennial elderly care policy plan followed by successive five-year plans. We scrutinized the evolution of China’s elderly care policy plans from 1994 to 2020. Based upon our results, we can identify six successive and often simultaneous strategic steps that China introduced to contend with the aging challenge.

At the first step, China heavily invested in the construction of elderly care institutions. This approach was not as successful as expected because a considerable number of homes remained vacant. The new facilities were often too costly and remote from family. Moreover, many elderly feared to lose face since moving to a nursing home could suggest that children are not caring enough [35].

The second step represents the new model of elderly care: home care as the cornerstone, community care as the backing and institutional care as the supplement (12th FYP) [64]. China’s deep-rooted tradition of taking care of older parents (filial piety) was eroding due to demographic and socioeconomic changes [65]. Western nations have experienced similar shifts since the 1960s replacing elderly institutional care by formal home care and community care services and then a recent switch to informal home-based care [12]. In nearly all European countries, the home-based care solutions, introduced for health, social and emotional well-being mainly aimed to contain public expenditure [12]. China’s third step is the strengthening of informal care with the uniqueness of the revival of filial piety. Since 2007, the Chinese government has promoted traditional Confucian values and is in the different FYPs requesting children to provide support to their older parents. The fourth step focuses on the paradigms of active and healthy aging. Active aging emerged in Europe two decades ago followed recently by ‘healthy aging’, which is promoted by the World Health Organization on the assumption that elderly ‘can provide significant social and economic benefits, in particular when they are active and healthy’ [65].

As a fifth step, marketization is gradually introduced in the FYPs. Different authors point to the risks of this strategy. According to Glinskaya and Feng [2], the concomitant effect will be the soaring of prices, which can affect China’s large middle class the most. The authors remark that underprivileged individuals are supported by public welfare and that the affluent population can afford care expenditures; however, the large middle class might not have easy access to the needed services without government subsidies.

The sixth step is the launching in 2017 of the ‘Action Plan for the Development of Smart Health and the Elderly Care Industry’. China is one of the leading countries in the development and implementation of smartphone apps; however, Chinese elder people seem still quite reluctant to use them. In order to develop an efficient care infrastructure, China is integrating high-tech in elderly care, such as virtual nursing and internet hospitals. Japan’s current technology development is leading toward ‘Society 5.0’ or ‘Super Smart Society’ going beyond technology, aiming at a prosperous human-centered society [66]. China’s recent 14th Economic Development Plan opens with the command: ‘insist on putting people at the center’ [25]. This policy heading toward 2035 is showing a similar orientation with a shift from high-speed development to a high-quality model [67].

The 7th step is the introduction of Long-Term Care Insurance (LTCI) in China. The Chinese government started to implement LTCI pilot regions from 2016 onwards, reaching a total of 49 pilots in 2020 [61]. Long-Term Care Insurance (LTCI) has become a general worldwide approach to financing the major costs of LTC services. LTCI will develop steadily in China. In fact, China has already shown its capability of advancing successfully in many other sectors. This trend is partly due to the latecomer advantage of developing countries acquiring and updating technologies and systems from advanced high-income countries [68]. Today, China is expanding on the LTCI market abroad and can extract the best elements of LTCI systems used elsewhere.

Although China’s elderly care policies represent unique characteristics, such as the community care structure and filial piety, they share common features found in Western approaches, such as the promotion of home-based care and healthy aging. In Japan, mutual aid among elderly and self-help have now the highest priority to alleviate the aging burden [69]. Steven Ney [70] argues that the political commitment to policies promoting the inclusion of the elderly in society has a rather rhetorical value. The governments are less pursuing social standpoints than financial objectives. Even though the cultural backgrounds of Western countries and Asian countries differ, elderly policies seem to converge. Similar as in Europe, the main burden of China’s elderly care is today put on the shoulders of the family and the elderly population themselves. The latter must try to stay healthy and strong to remain self-sufficient as long as possible. As Tu [71] points out, the ‘Care of the Self’ has turned into the ‘Entrepreneur of the Self’.

Our study has several limitations. First, we focused primarily on the successive perennial elderly policy plans at the national level. New studies should pay attention to the implementation of these plans at regional and city levels. Second, outcomes could not be verified, since the national plans only succinctly report achievements of previous plans. In addition, the policies were implemented according to local socioeconomic conditions, which we were not able to verify at this stage. This study, which provides a general overview of China’s national elderly policies, invites both more in-depth examination and field research on the topics discussed.

## 5. Conclusions

Despite the increase in income inequality, a considerable part of China’s population has experienced a real income rise. Elderly care policies show the willingness of China’s government to improve aging conditions. As in many other nations, substantial efforts are required from the elderly and their families themselves. In light of our study, we can conclude that although China is following a strategy with unique characteristics, elderly policies in China and Western countries are converging by emphasizing home-based care, informal caregivers and healthy aging.

## Figures and Tables

**Figure 1 ijerph-19-06141-f001:**
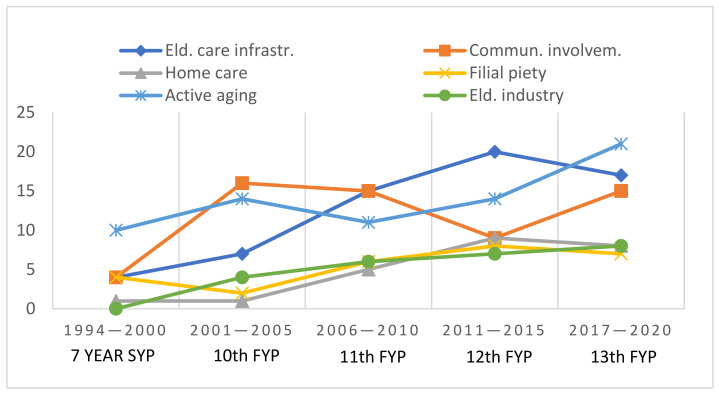
Frequency of attention on the six themes.

**Table 1 ijerph-19-06141-t001:** Highlights of China’s elderly care policies 1994–2020.

Policy Plan	Development Trend and Elderly Concern in the Respective Policies	Major Achievements
Seven Year Plan 1994–2000	Economic level: High economic growth Membership in World Trade Organization Domestic level: Economic development as solution for aging problem Elderly population: Increasing income disparity Set-up of nation-wide health care insurance and Pension system Basic welfare support First request for family tradition of supporting elderly First mention of support of elderly spiritual well-being First mention of active aging and self-support First mention of elderly to practice fitness First mention of protection of elderly rights and interests	1996: China’s Elderly Rights and Protection Law
10th Five-Year Plan 2001–2005	Economic level: High economic growth GDP growth rate in decade approximately 9.6% Worldwide major global player Domestic level: Accelerate science and technology Aging challenge will be put in overall plan of economy Introduce foreign capital in elderly care development Elderly population: Growing poverty rate among elderly Care services inaccessible and unaffordable Promote better care services and geriatric training Education for elderly also in preventive health care First mention of elderly self-reliance	2003: Fight against SARS epidemic and eradication 2004: Between 1981 and 2004, China lifted over half a billion people out of poverty
11th Five-Year Plan 2006–2010	Economic level: China maintains high rank in world market Domestic level: Social unrest State responds ‘care for entire population’ Major emphasis on welfare Use Public Welfare Lottery Fund for aged care First mention of elderly industry Elderly population: Inaccessible health care State responds with major health care reform Training of community nurses and GPs Set-up of geriatric facilities Special emphasis on family harmony Encourage to do sports and cultural activities	2006: The central government indicates new strategy to ‘put people first’ 2007: Confucian teachings and moral values are promulgated 2009: Major health care reform with primary care and hospital as key areas
12th Five-Year Plan 2011–2015	Economic level: Worldwide expansion Domestic level: New normal economic development: consumption driven, more service-oriented Social development and environmental protection Elderly population: Promulgation of new elderly care structure Emphasis on moral education, in particular on the filial piety virtue Promote elderly friendly cities and livable communities Encourage elderly to participate in economic development and volunteering	2011: Law on mandatory social insurances; Over 95% of the Chinese population is covered by health insurance; New elderly care structure: elderly home-based care as foundation; community elderly care as support; and state institutional care as supplement Goal: 30 beds per 1000 elderly
13th Five-Year Plan 2016–2020	Economic level: Belt and Road Initiative is expanding on international and China’s regional levels Domestic level: Lower GDP growth rate at 6.8% Still significant income disparity Urge elderly industry and marketization Elderly population: Moral values and filial piety must be honored Promote self-confidence of elderly At least 12% of the elderly population should engage in volunteering Support elderly to start own business	2016: The state launches plan “Healthy China 2030”, emphasizing care of chronic diseases 2017: Basic pension insurance participation rate reaches 90% The proportion paid by households out of-pocket declined from 60% in 2000 to 29% in 2017 Central government launches “Action Plan for the Development of Smart Health and the Elderly Care Industry (2017–2020)” 2020: Basic health insurance covers entire urban and rural population; over a billion people are protected by catastrophe medical insurance

## Data Availability

Not applicable.

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
