# Peer review of "China’s Elder Care Policies 1994–2020: A Narrative Document Analysis"

_ijerph, 2022, doi:10.3390/ijerph19106141_

Round 1

Reviewer 1 Report

Thank you very much for this interesting paper. This paper reviewed the evolution of elder care policy in China. Please find below my comments for further improvements of this manuscript:

General comment:

Please check grammatical errors in the thorough manuscript.

Introduction:

1) Please provide more background about elder people's situation in China, for example: statistics, future prediction, etc., then compare to other countries to illustrate the burden that China confronts.

Result

1) Please remove line 144-146

Discussion:

Overall, I feel this paper like a government report. I could find any discussion from authors' point of view about the evolution of the policies. Please elaborate and suggest implications after reviewing the policies.

Reviewer 2 Report

This is a review article to summarize the history and future directions of China's elder care systems. This is valuable, but it needs additional information and discussion. 

The authors described that they used a qualitative analysis, but it does not seem to be clear about how individual codes were grouped. To be more scientific, the author should show the process of qualitative analysis.

As for discussion, the authors did not compare the results with western countries. Individual Western countries have different background and cultures of elderly care. 

Round 2

Reviewer 2 Report

The paper has been revised in an appropriate manner. 

Author Response

We are very grateful for your approval of these revisions. Thanks very much for all your comments and suggestions on our paper.